# The Safety and Feasibility of Laparoscopic Surgery for Very Low Rectal Cancer: A Retrospective Analysis Based on a Single Center’s Experience

**DOI:** 10.3390/biomedicines9111720

**Published:** 2021-11-19

**Authors:** Hyuk-Jun Chung, Jun-Gi Kim, Hyung-Jin Kim, Hyeon-Min Cho, Bong-Hyeon Kye

**Affiliations:** 1Department of Surgery, Mulago National Referral Hospital, Kampala P.O. Box 7051, Uganda; thepursuit12@gmail.com; 2Department of Surgery, Pyeongtaek St. Mary’s Hospital, Pyeongtaek 17825, Korea; jgkim@catholic.ac.kr; 3Department of Surgery, Eunpyeong St. Mary’s Hospital, College of Medicine, The Catholic University of Korea, Seoul 03312, Korea; hj@catholic.ac.kr; 4Department of Surgery, St. Vincent’s Hospital, College of Medicine, The Catholic University of Korea, Suwon 16247, Korea; hmcho@catholic.ac.kr

**Keywords:** very low rectal cancer, sphincter saving surgery, laparoscopic surgery, oncologic outcomes

## Abstract

In this work we intend to validate the long-term oncologic outcomes for very low rectal cancer over the past 20 years and to determine whether laparoscopic procedures are useful options for very low rectal cancer. A total of 327 patients, who electively underwent laparoscopic rectal cancer surgery for a lesion within 5 cm from the anal verge, were enrolled in this study and their long-term outcomes were reviewed retrospectively. Of 327 patients, 70 patients underwent laparoscopic low anterior resection (LAR), 164 underwent laparoscopic abdominal transanal proctosigmoidocolectomy with coloanal anastomosis (LATA), and 93 underwent laparoscopic abdominoperineal resection (APR). The conversion rate was 1.22% (4/327). The overall postoperative morbidity rate was 26.30% (86/327). The 5-year disease free survival (DFS), 5-year overall survival (OS), and 3-year local recurrence (LR) were 64.3%, 79.7%, and 9.2%, respectively. The CRM involvement was a significant independent factor for DFS (*p* = 0.018) and OS (*p* = 0.042) in multivariate analysis. Laparoscopic APR showed poorer 5-year DFS (47.8%), 5-year OS (64.0%), and 3-year LR (17.6%) than laparoscopic LAR (74.1%, 86.4%, 1.9%) and laparoscopic LATA (69.2%, 83.6%, 9.2%). Laparoscopic procedures for very low rectal cancer including LAR, LATA, and APR could be good surgical options in selective patients with very low rectal cancer.

## 1. Introduction

After the introduction of abdominoperineal resection (APR) for rectal cancer treatment by Miles in 1908, APR was the standard procedure for all the rectal cancers located less than 5 cm from the anal verge. This was because at least 5 cm of distal margin was required up until the 1980s, after which 2 cm was considered adequate [1,2]. However, sphincter saving surgery for low rectal cancer has developed through the concept of total mesorectal excision (TME), which was introduced by Heald in 1982. TME enabled complete resection of rectal cancer and preservation of the pelvic autonomic nerves, awareness of the importance of the circumferential resection margin (CRM), acceptance of the distal resection margin (DRM) to be <1 cm or even <5 mm in terms of local recurrence (LR) or overall survival (OS) in patients with good risk tumors, and the concept of downsizing and downstaging from neoadjuvant chemoradiation therapy (nCRT) [3,4,5,6,7].

In the era of minimally invasive surgery (MIS), laparoscopic resection for colon cancer showed long-term oncologic safety, which was equivalent to that of open surgery for colon cancer in terms of local recurrence and overall survival [8,9,10,11]. After increasing the oncologic safety and popularity of laparoscopic resection for colon cancer, several large-scale multicenter randomized clinical trials (RCTs), such as the MRC-CLASSIC, COLOR II, and COREAN trials, showed no differences in the local recurrence or disease-free survival rate between laparoscopic and open surgery for rectal cancer [10,11,12,13,14]. In addition, although the initial results of the ACOSOG Z6051 and ALaCaRT trials failed to show noninferiority for pathologic outcomes in the laparoscopic resection group for rectal cancer compared with the open resection group [15,16]. Two-year disease free survival (DFS) and OS of laparoscopic rectal cancer surgery were not significantly different when compared to those of open surgery [17,18].

To the best of our knowledge, there is no study to validate the long-term oncologic outcome of laparoscopic surgery for very low rectal cancer (<5 cm) that only includes operating modalities such as laparoscopic low anterior resection (LAR), laparoscopic abdominal transanal proctosigmoidectomy with coloanal anastomosis (LATA), and laparoscopic APR. We started laparoscopic surgery for very low rectal cancer in April 1994 and first performed LATA in February 1996. Transanal abdominal transanal proctosigmoidectomy and coloanal anastomosis (TATA) was developed by Gerald Marks at the Thomas Jefferson University in 1984 [19]. In this study, we refer to the laparoscopic TATA as “LATA”.

Recently, various approaches to rectal cancer surgery, such as robotic surgery and transanal TME, have been introduced and widely implemented. Some reports have been published with the results of better postoperative outcomes and similar oncologic outcomes compared to laparoscopic rectal cancer surgery [20,21]. However, the surgical techniques of robotic surgery and transanal TME (taTME) can be based on laparoscopic surgery. Specifically, in most reports about robotic surgery and transanal TME, these approaches may be beneficial for patients with difficult rectal cancer, such as very narrow pelvis or very low rectal cancer [22]. Therefore, although laparoscopic surgery is not a standard approach for rectal cancer, laparoscopic surgery for difficult rectal cancer has to be evaluated as a reference surgery for MIS according to the specified difficulties.

In the present study, we intend to validate the long-term oncological outcome of laparoscopic surgery for very low rectal cancer over the past 20 years, including every consecutive case from the first case, and determine whether laparoscopic surgery for very low rectal cancer is an appropriate approach for very low-lying rectal cancer.

## 2. Materials and Methods

### 2.1. Patient Enrollment

We retrospectively collected data for 1231 patients who underwent curative resection for rectal cancer within 12 cm of the anal verge by a single colorectal surgeon between January 1994 and December 2016. Among these patients, the patients who underwent transanal local excision for T1 rectal cancer, total or subtotal proctocolectomy, combined resection of other organ, emergency operation including the Hartmann procedure, open surgery, and patients with rectal cancer located between 5 and 12 cm from anal verge, were excluded from this study. A total of 327 patients who electively underwent laparoscopic rectal cancer surgery for a lesion located within 5 cm from the anal verge were enrolled in this study (Figure 1).

### 2.2. Ethics

After obtaining review board approval from the Catholic University of Korea, CMC Clinical Research Coordination Center (VC21RASI0219), we analyzed the data and clinical information of these 327 patients.

### 2.3. Definition and Procedures

The rectum may be conveniently divided into thirds with its most proximal end at a variable level, which is usually several centimeters above the anterior peritoneal reflection, and its most distal end near the dentate line. The median length of an anal canal is 3 to 4 cm from the anal verge [23]. In the current study, we defined rectal cancer that was located <5 cm from the anal verge as very low rectal cancer [23,24,25]. We measured the tumor location for all patients with rigid proctosigmoidoscopy at the time of initial diagnosis. For all enrolled patients, we performed a total mesorectal excision (TME). We usually performed laparoscopic TME with five trocars. After the trocars were placed, the inferior mesenteric artery was ligated at the root and the left colon was mobilized by dissection between the mesocolon of the left colon and the retroperitoneum. After detachment of the left colon from the abdominal wall, splenic flexure mobilization was routinely performed for safe anastomosis. Moreover, TME was then performed while preserving the autonomic nervous system. During TME, the mesorectum was mobilized up to the level of the puborectalis muscle. If a distal resection margin (DRM) of 1 cm or greater from the tumor was attainable using a laparoscopic curvilinear stapler, a double stapling technique was applied to perform a low anterior resection (LAR). However, if this was not possible, a laparoscopic abdominal procedure was performed, followed by a transanal approach for coloanal anastomosis after changing the patient to the lithotomy position. In this technique, a circumferential incision for DRM was made around the dentate line to extract the mobilized colorectum through the anus. After the proximal resection margin (PRM) had been determined it was resected and a hand-sewn coloanal anastomosis was then performed. We defined the case of anastomosis performed by the transanal hand-sewn technique as LATA. Hence, intersphincteric resection was included into the LATA procedure. A diverting ileostomy or a colostomy was performed. Otherwise, APR was performed for rectal cancers with threatened circumferential resection margin (CRM) (including uncertainty of tumor invasion to the sphincter muscle or pelvic floor) on the preoperative imaging study or intraoperative finding. Our intention for radical rectal surgery was to secure a safe CRM and DRM. We performed LATA only in patients who had been expected to have a secure, grossly negative CRM and more than 1 cm of DRM in the operation room.

### 2.4. Staging Workup

In patients who had biopsy-proven adenocarcinoma in the very low-lying rectum, a colonoscopy was performed to search for a synchronous lesion and rigid proctosigmoidoscopy to measure the length between the lesion and the anal verge. For local staging, abdomen and pelvic computed tomography (CT), transanal ultrasound, and/or rectal magnetic resonance imaging (MRI) were used. In addition, abdomen and pelvic CT and chest CT or positron emission tomography-computed tomography (PET-CT) scans were obtained for staging workup to search for distant metastatic lesions, and serum carcinoembryonic antigen (CEA) levels were assayed after the lesion was confirmed to be an adenocarcinoma. In patients who underwent nCRT, the aforementioned imaging studies for staging workup were preoperatively repeated at 4 weeks after the end of nCRT.

### 2.5. Chemoradiation Therapy

Although the decision as to whether patients were treated by radical surgery following nCRT or radical surgery alone was dependent on the surgeon, and most patients who had clinical T3–T4 or N+ rectal cancer received nCRT with conventional fractionation as follows: 1.8 Gy per day; five fractions per week; and a total dose of 50.4 Gy/28 fractions (45 Gy/25 fractions initially to the whole pelvis, followed by 5.4 Gy/3 fractions as a boost to the gross tumor). All of the patients received two cycles of concurrent chemotherapy with radiotherapy [5-fluorouracil (5-FU), 400 mg/m^2^ (IV) 1 h before radiotherapy and leucovorin, 20 mg/m^2^ (IV) immediately before each dose of 5-FU on days 1–5 and 29–33]. Surgical treatment was performed within 6–8 weeks after the end of nCRT.

Neoadjuvant radiation was intentionally omitted for some relatively young female patients who had a plan for child bearing and in cases where the surgeon was sure to secure the safe circumferential resection margin and distal resection margin without nCRT. For the pathologic T3–T4 or N (+) tumors, which were not treated with nCRT, the intension was to treat them with postoperative chemoradiation therapy.

### 2.6. Follow Up

For all patients, follow-up data were obtained during routine clinical practices. Using abdomen and pelvic CT and chest CT or plain chest X-ray, patients were examined every 3 months during the first 2 years and then every 6 months for the rest of the 3-year to 5-year follow-up schedule. The cause and the date of death were obtained after examination of the medical records.

### 2.7. Primary Outcome

The primary outcome was the long-term oncologic outcome, which included overall survival (OS), disease-free survival (DFS), and local recurrence rate according to three operative modalities.

### 2.8. Secondary Outcome

The short-term perioperative outcomes, including overall postoperative morbidity and recovery course after surgery, were analyzed in the overall study population.

### 2.9. Data Collection

For evaluating patients’ preoperative condition, we analyzed sex, age, American Society of Anesthesiologist (ASA) score, body mass index, and serum preoperative CEA. Intraoperative parameters were analyzed with conversion, operation time, intraoperative blood loss, and whether or not there were intraoperative complications. For the postoperative short-term outcome, we compared the starting day of diet, postoperative hospital stay, and the severity of complications using the Clavien–Dindo classification among three surgical groups. We investigated the long-term oncological outcome with various factors, including pathologic findings.

### 2.10. Statistical Analysis

Continuous variables were compared using Student’s t-test and one-way ANOVA and were expressed as the mean ± standard deviation. Categorical variables were analyzed with the χ^2^ test and the Fisher’s exact test. The survival probability analysis was performed using the Kaplan–Meier method. The log-rank test was used to assess the difference of survival between strata. The Bonferroni correction was used for pairwise or multiple comparison. Significance was defined as a *p* value < 0.05. Multivariate analysis was applied with Cox’s proportional hazard regression model. By using forward stepwise selection, independent factors were analyzed and the statistically significant entry and staying values were set at 0.05. All statistical analyses were performed using the Statistical Package of the Social Sciences (SPSS) version 23 for Windows (IBM SPSS, Inc., Chicago, IL, USA).

## 3. Results

There were 201 men and 126 women in the present study. Table 1 shows the patients’ demographic findings according to three surgical techniques. The location of the tumor in the APR group was lower than those in the LAR or the LATA group (*p* < 0.001). In addition, nCRT was more frequently performed in the LATA or the APR group than in the LAR group (*p* = 0.002).

There was no postoperative mortality in the three groups. There was no conversion in the LAR group and there were two conversions in the LATA (1.2%) and APR (2.2%) groups, respectively, but there was no significance (*p* = 0.466). Urinary sequela was more common in the LATA and APR groups than in the LAR group (6.1% and 12.8% vs. 0%, *p* = 0.008). Postoperative hospital stay was longer in the APR group than in the LAR and the LATA groups (17.12 ± 20.24 days vs. 10.44 ± 6.22 and 12.59 ± 7.55 days, *p* = 0.002) (Table 2). Table 3 shows the list of early postoperative and late complications. The postoperative complications within 30 days after surgery occurred in 16 (22.9%) patients in the LAR group, 40 (24.7%) in the LATA group, and 30 (32.3%) in the APR group. Of these, the most common postoperative complication in all three operation groups was postoperative ileus, which occurred in 5 (7.1%) patients in the LAR group, 19 (11.6%) in the LATA group, and 12 (12.9%) in the APR group. Surgical site infections more frequently occurred in the APR group (13 patients, 14.0%, *p* < 0.001). The postoperative complications were significantly associated with being male (male vs. female; 30.7% vs. 19.8%, *p* = 0.031), low BMI (BMI ≤ 18.5 vs. BMI > 18.5; 47.1% vs. 25.2%, *p* = 0.047), nCRT (nCRT followed by TME vs. upfront surgery, 29.6%vs.8.7%, *p* = 0.008), advanced T stage (T1 vs. T2 vs. T3 vs. T4; 13.6% vs. 26.3% vs. 30.7%, vs. 39.1%, *p* = 0.033), and positive CRM (positive CRM vs. negative CRM; 44.0% vs. 24.9%, *p* = 0.038). The late complications that occurred after 30 days post operation were rectovaginal fistula (3 patients in the LAR group and 1 patient in the LATA group), anastomosis site stricture (7 patients in the LATA group), and stoma problems (3 patients in the LATA group and 2 patients in the APR group). In the present study, the diverting stoma was always performed during the LATA procedure. In all 140 patients in the LATA group, we were able to check if their diverting stomas were reversed or not. Seven (5.0%) of 140 patients did not undergo the operation for the reversal of their diverting stoma for following reasons: six patients refused the operation for the reversal of diverting stoma; one patient committed suicide due to neuropsychiatric problems before the reversal of the diverting stoma.

T4 cancer was more common in the APR group than in the LAR or LATA groups (15.1% vs. 4.3% and 3.7%, *p* = 0.004). However, there was no statistical difference in CRM involvement among the three groups. The PRM was shorter in the APR group than in the LAR or LATA groups (*p* = 0.004) and the DRM was shorter in the LATA group than in the LAR or APR groups (*p* < 0.001). There were 25 cases of R1 resection (they confirmed CRM involvement pathologically) and no R2 resection (Table 4).

The mean DFS and OS for all enrolled patients in the present study were 94.19 ± 3.5 and 110.74 ± 3.1 months. The 5-year DFS rate, 5-year OS rate, and 3-year local recurrence rate for all enrolled patients were 64.3%, 79.7%, and 9.2%, respectively (Figure 2). Table 5 shows the results of the oncological outcomes by univariate analysis according to various factors. The factors related to DFS were postoperative complication (*p* = 0.027), overall stage (*p* < 0.001), T stage (*p* < 0.001), lymph node metastasis (*p* < 0.001), CRM involvement (*p* < 0.001), histologic differentiation (*p* = 0.002), lymphatic invasion (*p* < 0.001), vascular invasion (*p* = 0.001), perineural invasion (*p* < 0.001), initial serum CEA level (*p* < 0.001), and postoperative serum CEA level (*p* < 0.001). The factors related to OS were intraoperative complication (*p* = 0.036), anastomotic leak (*p* = 0.005), overall stage (*p* < 0.001), T-stage (*p*= 0.001), lymph node metastasis (*p* = 0.003), CRM involvement (*p* = 0.038), histologic differentiation (*p* = 0.027), lymphatic invasion (*p* < 0.001), vascular invasion (*p* = 0.005), perineural invasion (*p* < 0.001), initial serum CEA level (*p* < 0.001), and postoperative serum CEA level (*p* < 0.001). In addition, the factors related to 3-year local recurrence were overall stage (*p* < 0.001), T and N stage (*p* < 0.001), CRM involvement (*p* = 0.001), lymphatic invasion (*p* = 0.001), perineural invasion (*p* = 0.003), initial serum CEA level (*p* = 0.048), and postoperative serum CEA level (*p* = 0.044). The surgical technique was a statistically significant factor in DFS (*p* = 0.001), OS (*p* = 0.001), and LR (*p* = 0.009) (Figure 3). In multivariate analysis, T stages (T0-1 and T3) and lymph node metastasis were significant factors in DFS (*p* = 0.002, *p* = 0.024, and *p* = 0.021, respectively). CRM involvement was a statistically significant factor in DFS (*p* = 0.018) and OS (*p* = 0.042). Postoperative CEA level was an independent factor in DFS and OS (*p* = 0.025 and *p* = 0.038) (Table 6).

## 4. Discussion

Operations for rectal cancer are technically more challenging than for colon cancer because the surgical fields for rectal surgery are confined by the narrow and deep pelvis and TME and autonomic nerve preservation are required for functional and oncological safety [12]. Laparoscopic surgery enables the surgeon to directly visualize the narrow pelvic cavity and to perform accurate and sharp dissections, while the magnified vision clearly delineates the anatomy, thus permitting a more precise TME [26]. Clinically, the way to find out the feasibility and safety of the surgical procedure is to check the short-term perioperative outcomes and oncologic outcomes in patients with cancers.

According to some reports for laparoscopic rectal cancer surgery, perioperative morbidity is about 20%–40% [27,28,29,30]. In the present study, the overall intraoperative and postoperative morbidity rate was 10.7% (35/327) and 26.3% (86/327), respectively. Of these, less surgical site infection might be one of the obvious advantages of laparoscopic rectal surgery compared with open surgery [28]. In the COREAN trial, the overall perioperative complication rate was 21.2% in the laparoscopic group and 23.5% in the open group [28]. They demonstrated that wound discharge, including seroma and superficial surgical site infection, were more common in the open group than in the laparoscopic group (1.2% vs. 6.5%, *p* = 0.020). In subgroup analysis of our study, the overall intraoperative or postoperative morbidity was not different among the three groups. However, surgical site infection was more frequently observed in the APR group than in the LAR or the LATA group (*p* < 0.001). Especially, in the LATA procedure only four or five small wounds within 1 cm on the abdomen are needed for trocar placement and the specimen is removed through the anus without any other wound for specimen extraction. On the other hand, in APR procedure a perineal wound is inevitable. In 2002, Poulin et al. reported that surgical site infection after laparoscopic TME mainly occurred in the perineal wound after laparoscopic APR. [26] In the present study, all of the surgical site infections (13/93, 14.0%) in the APR group were perineal wound infections. Many patients (82/93, 88.2%) in the APR group received preoperative nCRT, which made perineal wound healing difficult. These wound problems made the postoperative hospital stays longer in the APR group (*p* = 0.002). In South Korea, all patients are covered by a national insurance system, which makes hospital stays longer because patients are reluctant to be discharged when they feel any discomfort or any problem even if the problem is minor.

Most colorectal surgeons believe that the quality of TME may be one of the most important surrogate indicators for the oncologic outcome of rectal cancer surgery. The ACOSOG Z6051 and ALaCaRT trials did not show noninferiority of laparoscopic surgery compared with open surgery for rectal cancer for pathologic outcomes [15,16]. These trials had inconclusive results regarding the noninferiority of laparoscopy in terms of the quality of the surgical resection and further research is required to determine noninferiority. Hence, it does not mean laparoscopic resection is worse [31]. Both trials published 2-year DFS and LR data in 2019. In terms of mid-term oncologic outcomes, both trials reported similar oncologic outcomes in the laparoscopic and open surgery groups [17,18]. However, authors for both trials said lack of statistical difference was not an indicator of no difference existing and recommended the observation of longer term follow-up results to determine noninferiority of laparoscopic surgery for rectal cancer. Son et al. also recommended that the surgical community should be interested in the long-term outcomes of both trials because it was not certain whether near-complete TME has unfavorable oncologic impact and whether laparoscopic surgery with near-complete TME is an oncologic threat [32]. Cutis et al. showed that substantial variation in technical performance among credentialed surgeons can be seen and is significantly associated with clinical and pathological outcomes [33]. The authors stated that the upper-quartile-scoring surgeons obtained excellent results compared with the lower-quartile surgeons (mesorectal fascial plane: 93% vs. 59% *p* = 0.002; ALaCaRT composite end point success, 83% vs. 58% *p* = 0.03; 30-day morbidity, 23% vs. 50% *p* = 0.03) in the AlaCaRT trial. Unfortunately, in the present study, the quality of TME was not evaluated by a pathologist. However, the quality of laparoscopic rectal surgery can be evaluated by the rate of conversion to open and negative CRM. According to the landmark studies for laparoscopic rectal cancer surgery, the rate of conversion to open in the COREAN trial, COLOR II trial, MRC CLASICC trial, and Japanese cohort study are 1.2%, 16.6%, 34%, and 5.2%, respectively [12,27,28,29,30]. In addition, the rate of negative CRM in the laparoscopic group in the COREAN trial, COLOR II trial, MRC CLASICC trial, and Japanese cohort study were 97.1%, 93.0%, 84%, and 95.47%, respectively [27,28,29,30]. In the present study, the overall conversion rate was 1.22% and the rate of negative CRM in the present study was 92.35%. The rate of negative CRM in the LAR group, LATA group, and APR group were 94.2%, 93.3%, and 89.1%, respectively (*p* = 0.395). Comparing the conversion rate and the negative CRM rate with the other studies, our laparoscopic procedures might be also acceptable. In the present study, 5-year DFS rate, 5-year OS rate, and 3-year LR rate were 64.3%, 79.7%, and 9.2%, respectively. According to the landmark studies, the 3-year DFS in the laparoscopic group was 70–80% and the 3-year OS in the laparoscopic group was 86–91% [12,14,30]. The 3-year LR in the laparoscopic group was 2–10%. In addition, the long-term outcome of the 10-year follow-up of the COREAN trial was recently published. They reported that 5-year DFS rate, 5-year OS rate, and 5-year LR rate were 87.5%, 76.1%, and 2.5%, respectively and 10-year DFS rate, 10-year OS rate, and 10-year LR rate were 76.8%, 64.3%, and 3.4%, respectively [13]. Even though our study included stage I cancer, 77.8% of stage I cancer (112/144) was after nCRT. So the long-term oncologic outcomes are remarkable considering that only very low rectal cancers within AV 5 cm or less were included, and the results came from all consecutive laparoscopic surgeries for very low rectal cancers starting in 1994. In the COREAN trial, authors concluded a laparoscopic approach could be justified for rectal cancer surgery when performed by well-qualified colorectal surgeons. Since 1994, most rectal cancer patients have undergone laparoscopic surgery by one team composed of expert colorectal laparoscopic surgeons in South Korea. Meticulous sharp dissection following the exact surgical plane for TME and maintenance of oncologic principles for cancer surgery enabled good short-term and long-term oncologic outcomes. In the ACOSOG Z6051 trial, factors that negatively impacted DFS after the resection of rectal cancer were APR, low position of the tumor in the rectum, rectal perforation during the resection, and unsuccessful operation based on CRM positivity [17]. In the National Cancer Database (2004–2013) propensity matched analysis, patients undergoing LAR with coloanal anastomosis compared with APR for rectal cancer had better OS and were less likely to have positive margins despite the technically challenging operation [34]. The reasons for worse overall outcomes of APR mentioned in that study were the selection of more advanced tumors, which are more likely to perforate intraoperatively, inablility to obtain negative CRM with APR, and lymph node spread, which may not be adequately treated with APR. In our study, the APR group showed low tumor location (2.79 ± 1.31 cm, *p* < 0.001) and more T4 cancer (15.1%, *p* = 0.004) with a statistical significance. Those factors might impact worse long-term oncologic outcomes in the APR group. The CRM positive rate was higher in the APR group (10.9%) than in the LAR and LATA groups (5.8% and 6.7%) although it was not statistically significant (*p* = 0.395). Since CRM involvement was a significant factor for DFS and OS in multivariate analysis, a more aggressive approach with wider margins such as extra-levator APR or cylindrical resection is needed if APR is to be performed. According to our results, the LATA may be an oncologically safe surgical procedure with better DFS, OS, and LR than APR in case of technical feasibility.

For sphincter saving surgery for very low rectal cancer, it is important to have adequate distal resection margin microscopically as well as grossly from the anal sphincter. As we mentioned earlier, APR was the standard procedure for all the rectal cancers located below 5 cm from the anal verge because at least 5 cm of distal margin was required until the 1980s. In 1983, Pollett WG et al. reported a margin less than 2 cm below a rectal carcinoma did not affect survival or local recurrence adversely [2]. In 2001, Heidi Nelson et al. reported that for tumors of the distal rectum (<5 cm from the anal verge), the minimally acceptable length of the distal margin is 1 cm and this was supported by previous findings that subclinical distal bowel intramural spread is present within 1 cm distally from the visible tumor in a substantial proportion of patients [5,35]. In the present study, threatened DRM was defined as <1 cm DRM. The threatened DRM rate in the LATA group (32.3%) was higher than in the LAR (8.8%) or APR (7.7%) groups (*p* < 0.001). However, these differences were not correlated with the long-term oncologic outcomes. The threatened DRM was not significantly related to 5-year DFS (*p* = 0.816), 5-year OS (*p* = 0.885), or 3-year LR (*p* = 0.469) in univariate analysis and was not a significant factor in multivariate Cox regression analysis. A systematic review in 2012 supported the practice of sphincter preservation in selected settings of close distal margins (<1 cm) after TME for distal rectal cancer [5]. The review concluded that they could not find a statistically significant difference in either local control or survival with margins of <1 cm and margins as close as ≤5 mm—indeed negative—may be acceptable in patients with low-risk tumors or a good response to nCRT. In our study, the LATA was performed more frequently in males (65.2%) than females (34.8%) compared with the LAR (57.1% vs. 42.9%) and APR (58.1% vs. 41.9%) groups even though it was not significant statistically (***p*** = 0.369). The reason was the secure rectal transection maintaining the proper DRM and CRM with a stapling device, such as a laparoscopic GIA stapler, was more challenging in male patients than in female patients because of the narrower pelvic cavity in male patients. Consequently, more female patients were in the LAR group and more male patients were in the LATA group. When comparing the long-term oncologic outcome of the three surgical techniques, 5-year DFS (47.8%), 5-year OS (64.0%), and 3-year LR (17.6%) in the APR group showed significantly worse results than in the LAR (74.1%, 86.4%, 1.9%, respectively) and LATA (69.2%, 83.6%, 9.2%, respectively) groups (*p* = 0.001, *p* = 0.001, *p* = 0.009, respectively).

In recent years, surgical techniques have been developed and new technologies for rectal cancer surgery have been introduced. In particular, robotic TME and taTME are widely adopted and these approaches can yield good results [20,21,22]. Especially, robotic TME and taTME may be beneficial to patients with the difficult pelvis or very low rectal cancer. Park et al. demonstrated that the robotic surgery could make the surgeon perform meticulous and gentle dissection. They suggested that robotic surgery can lessen the technical difficulties and microscopic cancer cell dissemination and secure more resection margin in difficult cases [22]. The transanal phase of a taTME procedure can provide a clear view of the dissection plane low down in the pelvis with more cost efficiency than robotic or laparoscopic surgery [21]. Despite the application of robotic TME and taTME, the base of these techniques might be a laparoscopic surgery, which is still firstly faced MIS for most surgeons and has made the surgeons more familiar with magnified surgical views and instruments for MIS.

Our study has some limitations in that it was a retrospective study using the prospectively collected medical records and follow up was not able to be performed for all patients. Therefore, there were missing data. In addition, since this study focused on the possibility of laparoscopic surgery, risk factors for postoperative morbidity were not analyzed in detail. Our study was performed only in Korean populations from a single institution. There might be a reproducibility issue because our study only detailed the experience of a single expert surgeon. Since we included old data, we could not analyze TME completeness for quality and nCRT guidelines were not strictly applied to all indicated patients. Stages also included ypStages. Therefore, there are limitations for our results interpreted as generalized. In addition, although recently there has been a lot of interest in postoperative defecation function after sphincter preserving surgery, we could not analyze the functional outcome because large amounts of data were missing in our old data. Nevertheless, as far as we know, there is no paper that have analyzed all consecutive patients since the start for all possible procedures of laparoscopic surgery for very low rectal cancer (<5 cm from AV). Our analysis shows a good perspective for the short-term and long-term outcomes of laparoscopic surgery for very low rectal cancer.

## 5. Conclusions

Based on our findings, laparoscopic resection for very low rectal cancer shows acceptable short-term and long-term outcomes compared with previous landmark RCTs or a cohort study performed recently. Laparoscopic sphincter-saving procedures such as LAR and LATA could be good surgical options in selective patients with very low rectal cancer when performed by well-qualified colorectal surgeons. In addition, we suggest that laparoscopic rectal cancer surgery can serve as a reference surgery for rectal cancer patients in the context of advances in technology and surgical technique.

## Figures and Tables

**Figure 1 biomedicines-09-01720-f001:**
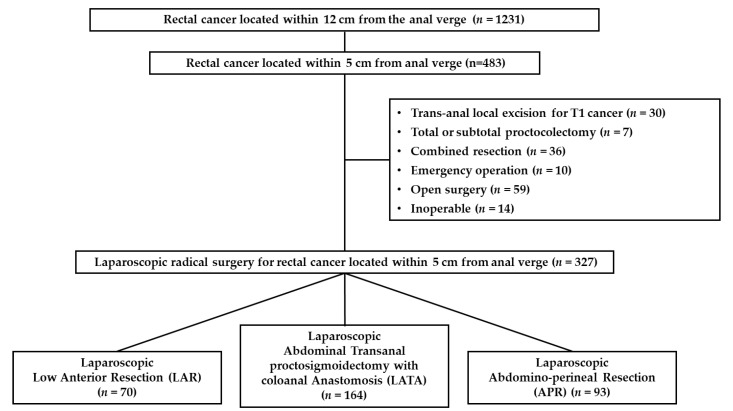
Flowchart of the inclusion process.

**Figure 2 biomedicines-09-01720-f002:**
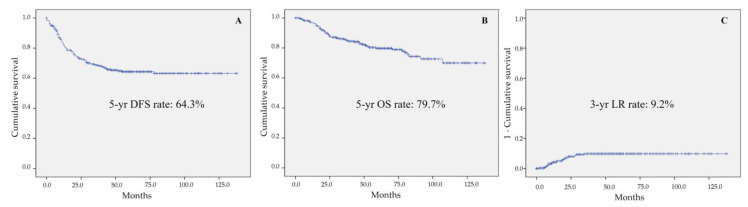
Cumulative survival of all enrolled patients: (**A**) 5-year disease free survival (DFS) rate, (**B**) overall survival (OS) rate, and (**C**) 3-year local recurrence (LR) rate.

**Figure 3 biomedicines-09-01720-f003:**
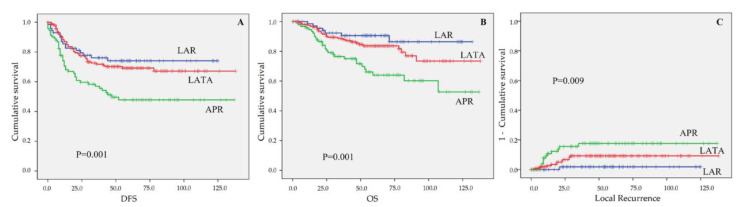
The oncologic outcome according to three operation techniques: (**A**) DFS, (**B**) OS, and (**C**) LR. (**A**) Mean DFS (LAR; 96.41 ± 6.1 months, LATA; 99.87 ± 4.7 months, APR; 74.34 ± 6.7 months) and DFS rate (at 5 years, LAR = 74.1%, LATA = 69.2%, APR = 47.8%, *p* = 0.001). (**B**) Mean OS (LAR; 119.02 ± 4.7 months, LATA; 114.24 ± 4.2 months, APR; 93.91 ± 6.4 months) and OS rate (at 5 years, LAR = 86.4%, LATA = 83.6%, APR = 64.0%, *p* = 0.001). (**C**) LR rate (at 3 years, LAR = 1.9%, LATA = 9.2%, APR = 17.6%, *p* = 0.009). DFS, disease-free survival; OS, overall survival; LR, local recurrence; LAR, low anterior resection; LATA, laparoscopic abdominal transanal proctosigmoidocolectomy with coloanal anastomosis; APR, abdominoperineal resection.

**Table 1 biomedicines-09-01720-t001:** Demographics of all enrolled patients.

	LAR (N = 70)	LATA (N = 164)	APR (N = 93)	*p*-Value
Age				
<65 years	38 (54.3%)	97 (59.1%)	55 (59.1%)	
≥65 years	32 (45.7%)	67 (40.9%)	38 (40.9%)	0.766
Mean ± SD	63.25 ± 10.98	61.31 ± 10.65	61.86 ± 11.86	0.468
Sex				
Male	40 (57.1%)	107 (65.2%)	54 (58.1%)	
Female	30 (42.9%)	57 (34.8%)	39 (41.9%)	0.369
BMI				
≤18.5 kg/m^2^	4 (5.8%)	9 (5.5%)	5 (5.4%)	
>18.5 kg/m^2^	65 (94.2%)	154 (94.5%)	87 (94.6%)	0.995
Mean ± SD	23.09 ± 3.41	23.73 ± 3.05	23.53 ± 3.22	0.377
ASA score				
1	32 (47.1%)	80 (49.7%)	50 (58.1%)	
2	33 (48.5%)	77 (47.8%)	33 (38.4%)	
3	3 (4.4%)	4 (2.5%)	3 (3.5%)	0.673
Location from AV (cm)	4.63 ± 0.66	3.79 ± 1.11	2.79 ± 1.31	<0.001
nCRT				
No	20 (28.6%)	15 (9.1%)	11 (11.8%)	
Yes	50 (71.4%)	149 (90.8%)	82 (88.2%)	0.002
Initial CEA				
≤5 ng/mL	46 (66.7%)	106 (67.5%)	48 (52.7%)	
>5 ng/mL	23 (33.3%)	51 (32.5%)	43 (47.3%)	0.053

LAR, low anterior resection; LATA, laparoscopic abdominal transanal proctosigmoidocolectomy with coloanal anastomosis; APR, abdominoperineal resection; SD, standard deviation; BMI, body mass index; ASA, American Society of Anesthesiologist; AV, anal verge; nCRT, neoadjuvant chemoradiation therapy; CEA, carcioembryonic antigen.

**Table 2 biomedicines-09-01720-t002:** Perioperative short-term outcomes and postoperative findings.

	LAR (N = 70)	LATA (N = 164)	APR (N = 93)	*p*-Value
Conversion	0	2 (1.2%)	2 (2.2%)	0.466
Intraoperative complication	4 (5.7%)	17 (10.4%)	14 (15.1%)	0.160
Postoperative complication	16 (22.9%)	40 (24.7%)	30 (32.3%)	0.311
Reoperation	2 (2.9%)	4 (2.4%)	1 (1.1%)	0.689
Urinary sequela	0	9 (6.1%)	10 (12.8%)	0.008
Postoperative hospital stay (days)	10.44 ± 6.22	12.59 ± 7.55	17.12 ± 20.24	0.002
Oral intake (POD)	4.94 ± 2.45	5.88 ± 3.74	5.95 ± 2.68	0.169
Postop CEA	64 (92.8%)	148 (92.5%)	68 (75.6%)	<0.001
≤5 ng/mL > 5 ng/mL	5 (7.2%)	12 (7.5%)	22 (24.3%)	

LAR, low anterior resection; LATA, laparoscopic abdominal transanal proctosigmoidocolectomy with coloanal anastomosis; APR, abdominoperineal resection; Intraop, intraoperative; Cx, complication; Postop, postoperative; POD, postoperative day.

**Table 3 biomedicines-09-01720-t003:** Postoperative complication in detail.

	LAR (N = 70)	LATA (N = 164)	APR (N = 93)	*p*-Value
Perioperative complications	16 (22.9%)	40 (24.7%)	30 (32.3%)	0.311
Postoperative ileus	5 (7.1%)	19 (11.6%)	12 (12.9%)	NS
Anastomosis leakage	2 (2.9%)	7 (4.3%)	0	NS
Postoperative bleeding	3 (4.3%)	2 (1.2%)	0	NS
Surgical site infection	2 (2.9%)	2 (1.2%)	13 (14.0%)	<0.001
Chylous ascites	2 (2.9%)	4 (2.4%)	2 (2.2%)	NS
Lung related	2 (2.9%)	1(0.6%)	1 (1.1%)	NS
Urinary tract related	1 (1.4%)	5 (3.0%)	6 (6.5%)	NS
C–D classification ≥3	5 (7.1%)	7 (4.3%)	0	0.047
Late complications	3 (4.3%)	11 (6.7%)	2 (2.2%)	NS
Anastomosis related Stoma problem	3 (4.3%) 0	8 (4.8%) 3 (1.8%)	0 2 (2.2%)	

LAR, low anterior resection; LATA, laparoscopic abdominal transanal proctosigmoidocolectomy with coloanal anastomosis; APR, abdominoperineal resection; C–D, Clavien–Dindo; NS, not significant.

**Table 4 biomedicines-09-01720-t004:** Comparison of pathologic results.

	LAR (N = 70)	LATA (N = 164)	APR (N = 93)	*p*-Value
Stage				
I	37 (52.9 %)	76 (46.3%)	31 (33.3%)	
II	21 (30.0%)	43 (26.2%)	32 (34.4%)	
III	12 (17.1%)	45 (27.4%)	30 (32.3%)	0.068
T stage				
0–1	16 (22.9%)	41 (25.0%)	11 (11.8%)	
2	25 (35.7%)	49 (29.9%)	25 (26.9%)	
3	26 (37.1%)	68 (41.5%)	43 (46.2%)	
4	3 (4.3%)	6 (3.7%)	14 (15.1%)	0.004
N stage				
0	57 (82.6%)	116 (72.5%)	62 (67.4%)	
1 or 2	12 (17.4%)	44 (27.5%)	30 (32.6%)	0.094
PRM				
<10 cm	5 (7.8%)	6 (3.9%)	14 (16.3%)	
≥10 cm	59 (92.2%)	148 (96.1%)	72 (83.7%)	0.004
DRM				
<1 cm	6 (8.8%)	51 (32.3%)	7 (7.7%)	
≥1 cm	62 (91.2%)	107 (67.7%)	84 (92.3%)	<0.001
CRM involvement *				
negative	65 (94.2%)	152 (93.3%)	82 (89.1%)	
positive	4 (5.8%)	11 (6.7%)	10 (10.9%)	0.395
Differentiation				
Well	19 (27.1%)	40 (24.4%)	16 (17.2%)	
Moderately	44 (62.9%)	101 (61.6%)	63 (67.7%)	
Poorly	0	5 (3.0%)	7 (7.5%)	0.142
Lymphatic invasion				
Yes	10 (14.3%)	28 (17.1%)	25 (26.9%)	
No	52 (74.3%)	111 (67.7%)	56 (60.2%)	0.206
Venous invasion				
Yes	2 (2.9%)	2 (1.2%)	6 (6.5%)	
No	61 (87.1%)	136 (82.9%)	77 (82.8%)	0.121
Perineural invasion				
Yes	5 (7.1%)	16 (9.8%)	19 (20.54)	
No	57 (81.4%)	123 (75.0%)	64 (68.8%)	0.052

LAR, low anterior resection; LATA, laparoscopic abdominal transanal proctosigmoidocolectomy with coloanal anastomosis; APR, abdominoperineal resection; PRM, proximal resection margin; DRM, distal resection margin; CRM, circumferential resection margin * CRM involvement negative: ≥ 1 mm, positive: < 1 mm.

**Table 5 biomedicines-09-01720-t005:** The results of oncologic outcomes according to various factors through the univariate analysis.

	5-Year DFS (%)	*p*-Value	5-Year OS (%)	*p*-Value	3-Year LR (%)	*p*-Value
Sex						
Male	63.5		79.4		9.8	
Female	65.6	0.669	80.4	0.931	9.9	0.966
Age						
<65 years	65.0		78.5		12.6	
≥65 years	63.0	0.977	81.4	0.837	6.0	0.068
ASA						
I	66.8		81.4		9.7	
II	63.4	0.595	79.1	0.460	8.8	0.401
III	42.9		87.5		25.0	
nCRT						
No	78.2		87.1		2.3	
Yes	62.1	0.094	78.3	0.735	11.4	0.226
Intraop Cx						
No	64.5		81.6		9.6	
Yes	64.4	0.743	66.5	0.036	12.7	0.639
Postop Cx						
No	68.1		82.4		8.1	
Yes	52.4	0.027	71.0	0.137	15.7	0.144
Anastomosis leak						
No	64.3		80.4		9.9	
Yes	46.9	0.290	46.7	0.005	25.0	0.555
Stage						
I	83.4		92.5		2.5	
II	57.2	<0.001	73.3	<0.001	10.0	<0.001
III	41.8		65.2		24.9	
T stage						
0–1	90.6		91.6		0	
2	73.5		89.4		7.2	
3	47.8	<0.001	68.4	0.001	14.2	<0.001
4	44.5		69.0		33.5	
N stage						
N0	72.0		81.4		5.8	
N(+)	42.8	<0.001	56.3	0.003	30.2	<0.001
PRM						
<10 cm	65.9		75.0		11.2	
≥10 cm	62.3	0.730	79.8	0.171	9.8	0.890
DRM						
<1 cm	63.4		76.6		11.9	
≥1 cm	63.9	0.816	80.2	0.885	9.4	0.469
CRM involvement						
(+)	34.9		59.9		27.3	
(−)	66.9	<0.001	81.5	0.038	8.7	0.001
Differentiation						
Well	77.1		88.8		5.7	
Moderately	57.4	0.002	74.3	0.027	6.2	0.336
Poorly	50.0		64.8		12.4	
Lymphatic inv						
(+)	42.0		62.7		23.5	
(−)	67.4	<0.001	85.3	<0.001	8.7	0.001
Venous inv						
(+)	45.0		53.3		0	
(−)	62.2	0.001	80.5	0.005	11.7	0.081
Perineural inv						
(+)	32.3		59.4		23.8	
(−)	66.6	<0.001	82.9	<0.001	9.8	0.003
Initial CEA						
≤5 ng/mL	73.1		88.1		7.2	
>5 ng/mL	47.8	<0.001	66.6	<0.001	15.4	0.048
Postop CEA						
≤5 ng/mL	70.4		88.4		9.1	
>5 ng/mL	16.4	<0.001	21.9	<0.001	15.7	0.044
Operation techniques						
LAR	74.1		86.4		1.9	
LATA	69.2	0.001	83.6	0.001	9.2	0.009
APR	47.8		64.0		17.6	

DFS, disease-free survival; OS, overall survival; LR, local recurrence; inv, invasion; LAR, low anterior resection; LATA, laparoscopic abdominal transanal proctosigmoidocolectomy with coloanal anastomosis; APR, abdominoperineal resection.

**Table 6 biomedicines-09-01720-t006:** The results of oncologic outcomes according to significant factors through the multivariate Cox regression analysis.

	DFS			OS		
	HR	*p*-Value	95% CI	HR	*p*-Value	95% CI
T0, Tis, T1	1	0.002		1	0.061	
T2	2.009	0.263	0.592∼6.819	1.294	0.744	0.275∼6.097
T3	3.917	0.024	1.196∼12.831	2.157	0.314	0.483∼9.623
T4	1.367	0.684	0.304∼6.145	0.350	0.350	0.039∼3.161
LN metastasis	1.742	0.021	1.088∼2.789	1.389	0.373	0.674∼2.865
CRM involvement	2.431	0.018	1.168∼5.062	2.839	0.042	1.038∼7.768
Lymphatic Invasion	1.176	0.555	0.686∼2.018	1.726	0.175	0.785∼3.795
Vascular Invasion	1.725	0.259	0.670∼4.445	2.791	0.103	0.811∼9.604
Perineural Invasion	1.580	0.113	0.898∼2.779	1.936	0.096	0.889∼4.218
Preop CEA	1.002	0.457	0.997∼1.008	0.997	0.604	0.987∼1.008
Postop CEA	1.012	0.025	1.002∼1.023	1.017	0.038	1.001∼1.034
Operation technique						
LAR	1	0.209		1	0.311	
APR	1.657	0.116	0.883∼3.108	2.228	0.130	0.789∼6.294
LATA	1.170	0.609	0.641∼2.138	1.696	0.303	0.621∼4.631

HR, harzard ratio; CI, confidence interval, LN, lymph node; CRM, circumferential resection margin; CEA, carcinoembryonic antigen; LAR, low anterior resection; LATA, laparoscopic abdominal transanal proctosigmoidocolectomy with coloanal anastomosis; APR, abdominoperineal resection.

## Data Availability

The datasets during and/or analyzed during the current study are available from the corresponding author on reasonable request.

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
