# Peer review of "The Safety and Feasibility of Laparoscopic Surgery for Very Low Rectal Cancer: A Retrospective Analysis Based on a Single Center’s Experience"

_biomedicines, 2021, doi:10.3390/biomedicines9111720_

Round 1

Reviewer 1 Report

The authors intended to validate the long-term oncologic outcomes for very low rectal cancer over the past 20 years and to determine laparoscopic procedures are useful options for very low rectal cancer. A total of 327 patients who electively underwent laparoscopic rectal cancer surgery for a lesion within 5cm from the anal verge were enrolled in this study and their long-term outcomes were reviewed retrospectively. Of 327 patients, 70 patients underwent laparoscopic low anterior resection (LAR), 164 laparoscopic abdominal transanal proctosigmoidocolectomy with coloanal anastomosis (LATA), 93 laparoscopic abdominoperineal resection (APR). The conversion rate was 1.22% (4/327). The overall postoperative morbidity rate was 26.30% (86/327). The 5-year disease free survival (DFS), 5-year overall survival (OS), and 3-year local recurrence (LR) were 64.3%, 79.7%, and 9.2%, respectively. The CRM involvement was significant independent factor for DFS (P=0.018) and OS (P=0.042) in multivariate analysis. Laparoscopic APR showed poorer 5-year DFS (47.8%), 5-year OS (64.0%), 3yr LR (17.6%) than laparoscopic LAR (74.1%, 86.4%, 1.9%) and laparoscopic LATA (69.2%, 83.6%, 9.2%). Laparoscopic procedures for very low rectal cancer including LAR, LATA, and APR could be good surgical options in selective patients with very low rectal cancer.

However there are some minor critiques that needs to be addressed

  1. Do the authors analyzed the risk factors for post-operative morbidity?
  2. Please include a schematic representation of the entire study as a separate figure.

Author Response

  1. Do the authors analyzed the risk factors for post-operative morbidity?
  • I added the result for the risk factors associated with postoperative complications in the Result section.
  1. Please include a schematic representation of the entire study as a separate figure.
  • I added a schematic figure into the Method section.

Thank you for your nice review and kind comment. I added some results for the risk factors associated with postoperative complications and a schematic figure of the entire study.

Reviewer 2 Report

The paper is a single surgeon retrospective case review of 327 patients undergoing surgery for very low rectal cancer. The oncolgy outcomes and the small number of post operative complications are impressive.

Case note review of consecutive cases does not rank highly in the level of medical evidence, but sometimes such figures can be useful to furnish a wider surgical debate. At present the surgical debate for very low rectal cancers is around technique (robotic or TaTME) and post operative function (low anterior resection syndrome).

This paper presents laparoscopic technical results which will provide a useful comparator/ benchmark for robotic and TaTME papers - as such it should be considered for publication.

The major omission from my point of view is the lack of postoperative functional data. Low anterior resection syndrome is increasingly recognised and debated in the surgical literature. There is a growing opinion that APR might be a more favourable operation in terms of quality of life over low anterior resection, particularly after neoadjuvant radiotherapy in certain sub populations eg frail, elderly, and poor pre-operative continence. The authors make no referrence to function in their paper - this should be addressed.

Author Response

The major omission from my point of view is the lack of postoperative functional data. Low anterior resection syndrome is increasingly recognised and debated in the surgical literature. There is a growing opinion that APR might be a more favourable operation in terms of quality of life over low anterior resection, particularly after neoadjuvant radiotherapy in certain sub populations eg frail, elderly, and poor pre-operative continence. The authors make no referrence to function in their paper - this should be addressed.

  • Thank you for your nice review and valuable comments. I agree with you. The functional outcome related to defecation like LARS is very important to the patients who undergo rectal cancer surgery. Unfortunately, this study is designed in a retrospective nature. Thus, we could not obtain information about defecation dysfunction from patients who underwent radical surgery before 2010. So, the information related to the defecation problem from lots of patients was missing. Since 2011, we have been trying to present our data related to postoperative functional outcomes. Recently, our department published our data about postoperative bowel dysfunction like LARS (Effect of Biofeedback Therapy during Temporary Stoma Period in Rectal Cancer Patients: A Prospective Randomized Trial. Hyeon-Min Cho, Hyungjin Kim, RiNa Yoo, Gun Kim, and Bong-Hyeon Kye. J. Clin. Med. 2021, 10(21), 5172; doi:10.3390/jcm10215172). Currently, we are conducting some studies about postoperative functional outcomes. In the near future, I am looking forward to presenting our results to the readers. However, I would like to say that I am very sorry for not presenting postoperative functional outcomes in this manuscript due to lots of missing data.
  • I added a sentence into the limitation in Discussion section.
  • Once again, I greatly appreciate your nice review and comments.

This manuscript is a resubmission of an earlier submission. The following is a list of the peer review reports and author responses from that submission.

Round 1

Reviewer 1 Report

The safety and feasibility of laparoscopic surgery for very low rectal cancer: A retrospective analysis based on a single center experience

In this study, the authors validated the long-term oncologic outcomes for very low rectal cancer over the past 20 years and to determine laparoscopic sphincter saving procedures are useful options for very low rectal cancer However, there are some critiques that need to be addressed.

  • In the Table 1 demographics data please include the diabetes, smoking habit and alcohol habit.
  • What is the percentage of operative mortality in these patients?
  • What are the other post-operative complications such as renal failure or cardiac complications observed in these patients? Please include few lines in the results.
  • Please include a table for operative details if possible.

Author Response

In this study, the authors validated the long-term oncologic outcomes for very low rectal cancer over the past 20 years and to determine laparoscopic sphincter saving procedures are useful options for very low rectal cancer However, there are some critiques that need to be addressed.

  • In the Table 1 demographics data please include the diabetes, smoking habit and alcohol habit.

--> Thank you for your nice review and suggestion. I agree with your opinion that the DM, smoking, and drinking habit are risk factors for postoperative morbidities. However, this study had a limitation of retrospective study. At this point, I have no data for these risk factors. I added this content into our study limitation in Discussion section.

  • What is the percentage of operative mortality in these patients?

--> There was no postoperative mortality in three groups. So, I added this sentence into Results section.

  • What are the other post-operative complications such as renal failure or cardiac complications observed in these patients? Please include few lines in the results.

--> Thank you for your valuable comments. Despite reviewing the data, no complications such as renal failure or heart problems were found. However, I did find complications of the kind with a small incidence in our data. So I added some complications to Table 3

  • .Please include a table for operative details if possible.

--> We added some sentences about operative details into Method section.

Once again, I really appreciate your nice and kind review. I am looking forward to your positive decision for this study.

Thank you.

Reviewer 2 Report

The authors present a retrospective  paper, but the manuscript needs to be improved.

The authors collect data between 1994 and 2011, why only between that date, for an update of the study it would be advisable to provide more current data.

Introduction, needs to be updated, where the problem/background of the study is framed and the hypothesis and objective are clear. It needs to be updated with recent literatura

Discussion; needs to be restructured, the authors restate their results. This section needs an interpretation of the results and comparison with existing studies, especially authoritative studies. Unnecessary paragraphs that are already stated in the results section should be removed and the focus should be on the discussion of the results.

The studies referenced by the authors are old. An update of the bibliography is needed for both the introduction and discussion sections

Author Response

The authors present a retrospective  paper, but the manuscript needs to be improved.

The authors collect data between 1994 and 2011, why only between that date, for an update of the study it would be advisable to provide more current data.

--> Thank you for your kind and good review. I totally agree with your comments. However, since 2011, there have been some changes in our hospital's medical record system. Since there was a lot of missing data for about 3-4 years after 2011, we set this interval between 1994 and 2011 for a more accurate and objective analysis. We are working to recover data from that period and plan to do a larger retrospective analysis with complete laparoscopic surgery data in the near future.

Introduction, needs to be updated, where the problem/background of the study is framed and the hypothesis and objective are clear. It needs to be updated with recent literatura

--> Thank you for your nice comments. I added some recent literatures into Introduction section.

Discussion; needs to be restructured, the authors restate their results. This section needs an interpretation of the results and comparison with existing studies, especially authoritative studies. Unnecessary paragraphs that are already stated in the results section should be removed and the focus should be on the discussion of the results.

--> Thank you for your nice comments. I tried to revise the Discussion as you comment.

The studies referenced by the authors are old. An update of the bibliography is needed for both the introduction and discussion sections

--> Thank you for your comments. I upgraded references and added more recently published ones into the Introduction and Discussion sections.

Once again, I really appreciate your nice and kind review. I am looking forward to your positive decision for this study.

Thank you.

Round 2

Reviewer 2 Report

The authors present a version of the manuscript similar to the previous one. The changes noted above have not been reflected. 
The introduction and discussion need to be expanded. In the introduction contextualise the study and in the discussion analyse the results, especially with recent literature. Five references have been added, of which only one is the most current.
In addition, the data provided are somewhat outdated.

Author Response

The authors present a version of the manuscript similar to the previous one. The changes noted above have not been reflected.

The introduction and discussion need to be expanded. In the introduction contextualise the study and in the discussion analyse the results, especially with recent literature. Five references have been added, of which only one is the most current.

In addition, the data provided are somewhat outdated.

  • Thank you for your nice review. I tried to revise the Introduction and Discussion as you comment. Our study was single center study. So, I know this study has lots of limitations. Therefore, we mainly intended to compare with historical landmark studies, especially authoritative studies like CLASSIC, COLOR II, COREAN, JAPANESE Cohort which shows short term and long term oncologic outcome of similar periods with our study.

          Once again, I really appreciate your nice and kind review. I am looking         forward to your positive decision for this study.

Thank you.

Round 3

Reviewer 2 Report

Although the introduction and discussion have been improved, the study has many limitations, which the authors themselves reflect throughout the manuscript. 
The authors themselves indicate the lack of a clear conclusion to the aim of their study